# Feasibility of Combining Transcranial Direct Current Stimulation and Active Fully Embodied Virtual Reality for Visual Height Intolerance: A Double-Blind Randomized Controlled Study

**DOI:** 10.3390/jcm11020345

**Published:** 2022-01-11

**Authors:** Samuel Bulteau, Andrew Laurin, Kalyane Bach-Ngohou, Morgane Péré, Marie-Anne Vibet, Jean-Benoit Hardouin, Véronique Sebille, Lydie Lagalice, Élodie Faurel-Paul, Didier Acier, Thomas Rabeyron, Valéry-Pierre Riche, Anne Sauvaget, Florian Melki, Toinon Vigier, Matthieu Perreira Da Silva, Olivier Charlet, Yannick Prié

**Affiliations:** 1Department of Addictology and Consultation-Liaison Psychiatry, CHU Nantes, F-44000 Nantes, France; andrew.laurin@chu-nantes.fr (A.L.); lydie.lagalice@chu-nantes.fr (L.L.); anne.sauvaget@chu-nantes.fr (A.S.); 2French Institute of Health and Medical Research (INSERM), UMR 1246 SPHERE, Université of Nantes, F-44000 Nantes, France; jean-benoit.hardouin@univ-nantes.fr (J.-B.H.); veronique.sebille@univ-nantes.fr (V.S.); 3Laboratory “Movement, Interactions, Performance” (MIP), University of Nantes, E.A. 4334, F-44000 Nantes, France; 4The Enteric Nervous System in Gut and Brain Disorders, INSERM, IMAD, Department of Biology, Laboratory of Clinical Biochemistry, and TENS, CHU Nantes, F-44000 Nantes, France; kalyane.bach@chu-nantes.fr; 5Department of Methodology and Biostatistics, CHU Nantes, F-44000 Nantes, France; morgane.pere@chu-nantes.fr (M.P.); marieanne.vibet@chu-nantes.fr (M.-A.V.); 6Department of Promotion, CHU Nantes, F-44000 Nantes, France; elodie.faurelpaul@chu-nantes.fr; 7Department of Psychology, University of Nantes, F-44312 Nantes, France; didier.acier@univ-nantes.fr; 8Department of Psychology, University of Lorraine, Interpsy, F-54000 Nancy, France; thomas.rabeyron@univ-lorraine.fr (T.R.); olivier.charlet.psychologue@gmail.com (O.C.); 9Economic Evaluation and Health Products Development Service, Partnership and Innovation Department, CHU Nantes, F-44000 Nantes, France; valerypierre.riche@chu-nantes.fr; 10LS2N—UMR 6004 CNRS, University of Nantes, F-44000 Nantes, France; florian.melki@univ-nantes.fr (F.M.); toinon.Vigier@univ-nantes.fr (T.V.); matthieu.perreiradasilva@univ-nantes.fr (M.P.D.S.); yannick.prie@univ-nantes.fr (Y.P.)

**Keywords:** virtual reality, transcranial direct current stimulation, anxiety, visual height intolerance, therapy, treatment

## Abstract

Background: Transcranial Direct Current Stimulation (tDCS) and Virtual Reality Exposure Therapy (VRET) are individually increasingly used in psychiatric research. Objective/Hypothesis: Our study aimed to investigate the feasibility of combining tDCS and wireless 360° full immersive active and embodied VRET to reduce height-induced anxiety. Methods: We carried out a pilot randomized, double-blind, controlled study associating VRET (two 20 min sessions with a 48 h interval, during which, participants had to cross a plank at rising heights in a building in construction) with online tDCS (targeting the ventromedial prefrontal cortex) in 28 participants. The primary outcomes were the sense of presence level and the tolerability. The secondary outcomes were the anxiety level (Subjective Unit of Discomfort) and the salivary cortisol concentration. Results: We confirmed the feasibility of the association between tDCS and fully embodied VRET associated with a good sense of presence without noticeable adverse effects. In both groups, a significant reduction in the fear of height was observed after two sessions, with only a small effect size of add-on tDCS (0.1) according to the SUD. The variations of cortisol concentration differed in the tDCS and sham groups. Conclusion: Our study confirmed the feasibility of the association between wireless online tDCS and active, fully embodied VRET. The optimal tDCS paradigm remains to be determined in this context to increase effect size and then adequately power future clinical studies assessing synergies between both techniques.

## 1. Introduction

Anxiety disorders affect about one third of the general population and are associated with significant distress and impairment due to excessive and persistent fear and/or anxiety and related behavioral disturbance, including avoidance mechanisms [1,2,3]. Pharmacological and cognitive behavioral therapies have proved their efficacy [4,5], but only half of patients respond [6,7], and one-third may discontinue their medication [8]. Anxiety disorders are associated with an economic burden [9], reaching more than 74 billion euros per year in Europe in 2010 [10]. Among anxiety disorders, acrophobia is one of the most common phobias, with a prevalence of 3 to 5% in the general population [11]. Visual Height Intolerance (vHI) corresponds to a significant fear of height without panic attacks and systematic avoidance [11]. vHI affects up to 28% of the general population, in which half present a significant decrease in quality of life [11,12].

Exposure therapy is based on fear extinction learning, enabling patients’ anxiety to be decreased [13,14,15]. In this regard, the efficacy of virtual reality exposure therapy (VRET), which consists of immersing the subject in a computer-generated virtual environment, has shown promise in treating a variety of anxiety disorders [16,17] and may be as effective as in vivo exposure therapy for patients suffering from acrophobia [18]. VRET has the advantage of saving time and costs and being suitable for patients who are too anxious to undergo in vivo exposure, with a transferable effect to real-life situations [19]. Moreover, successful attempts to enhance the effects of VRET have been carried out using drugs such as D-Cycloserine [20], while the literature about the interest of add-on non-invasive brain stimulation (NIBS) techniques is still scarce.

Studies investigating the pathophysiology of anxiety disorders have revealed that the ventral medial prefrontal cortex (vmPFC) is implicated in the process of fear extinction learning [21,22,23] throughout amygdala hyperactivity inhibition [24]. Furthermore, a recent study reported that anxiety intensity was correlated to vmPFC hypo-metabolism [25]. NIBS techniques are more and more often used to treat a variety of neuropsychiatric conditions [26,27], and are increasingly studied in relation to anxiety disorders [28,29]. Interestingly, the stimulation of the vmPFC using repetitive transcranial magnetic stimulation (rTMS) before virtual reality (VR) exposure (offline stimulation) in patients affected by acrophobia recently proved to enhance exposure therapy efficiency [30].

Transcranial direct current stimulation (tDCS) has shown its safety in healthy subjects and patients [31,32,33] and its ability to improve conditioned fear extinction [34,35,36]. This portable, easy-to-use device is cheaper than rTMS [37] and allows concomitant stimulation and behavioral therapy (i.e., *online* stimulation). Thus, tDCS could enhance neuronal plasticity given its neuromodulatory effects [38] and allow the optimization of the effects of tDCS based on the dependency of the neuronal network’s state while the subjects engage in a specific task (state-dependency) [39,40,41]. For example, *online* tDCS applied on the vmPFC in healthy individuals modulates emotional reactivity during exposure to affective stimuli [42].

Several studies have assessed the association between tDCS and VR, especially in neurology for cerebral palsy or stroke rehabilitation [43], but these studies used semi-immersive paradigms. To date, only two study assessed VR associated with tDCS in psychiatry: an exploratory study about impulsivity in 2008 [44] and more recently, an open-label pilot study involving 12 subjects in relation to post-traumatic stress disorder that suggested a reduction in stress reactivity according to galvanic skin reactivity measurements over time after concomitant passive immersion in war scenes and left vmPFC anodal stimulation [36]. Given these preliminary results, more studies are needed to identify adequate methodology that can be used to combine tDCS and immersive exposition eliciting fear extinction.

We hypothesized that it may be technically feasible to associate wireless online tDCS with VRET for vHI using an active, fully embodied and immersive virtual task. Our primary outcomes for feasibility were the sense of presence, cybersickness and dropout rates. The second aim of this initial experiment was to describe the efficacy (and to understand the effect size with this particular experimental setting) of the virtual exposure with and without add-on tDCS in reducing height-induced anxiety.

## 2. Material and Methods

### 2.1. Design

This study was conducted using a randomized, double-blind and controlled between-subjects design comparing the effect of two sessions of VRET combined with active or sham tDCS. The study was pre-registered in ClinicalTrials.gov (Identifier NCT03387254, last update date 30 July 2018.).

The study protocol included (Figure 1):Two assessment sessions measuring SUD during a virtual reality task lasting 10 min to measure participants’ fear of height at baseline (V1) and after VRET (V4).Two exposure sessions (V2 and V3), during which participants were exposed to height (VRET) for 20 min while being stimulated *online* (active tDCS) or not (sham tDCS). Two sessions is the minimal number of sessions needed to observe VRET efficacy in the literature. We chose that number since the aim of research regarding potentiating the VRET effect with other interventions is to achieve a stronger and quicker effect [20].

V1, V2, V3 and V4 took place on Mondays, Tuesdays, Wednesdays and Fridays, respectively, to allow 48 h between VRET sessions and 24 h between V3 and V4 to allow consolidation in the fear extinction neurobiological process (Figure 2: flow-chart).

### 2.2. Participants

Twenty-eight volunteers were recruited through advertisement at the University of Nantes as well as in radio, local newspapers and social media. They were considered healthy—they did not have any major psychiatric disorders, notably anxiety disorders, mood disorders, psychotic symptoms or substance use disorders, and they did not take any psychotropic treatment—according to their statement on a web-based screening questionnaire.

The inclusion criteria were a sufficient level in the *Acrophobia Questionnaire* (AQ) with a total score ≥ 45 [45] associated with a *Visual Height Intolerance Severity Scale* (vHISS) score ≥ 7 [12] and a *Subjective Unit of Distress* (SUD) rating ≥ 5/10 [46] at baseline (V1). A SUD score ≥ 5—with an AQ score ≥ 45 and a vHISS score ≥ 7—at baseline was required to ensure that the patient had enough anxiety at height exposure in concordance with the existing literature to adequately assess the potential effects of VRET associated with tDCS to reduce it.

The exclusion criteria were: DSM-5 acrophobia diagnosis (including avoidance and panic attack criteria); tDCS contraindication (personal history of neurosurgery, cephalic medical implant or cutaneous hyper sensibility); personal history of psychiatric or addictive disorder, psychotropic intake, intake of treatments that notably influence one’s mood or anxiety, neurological pathology, locomotor or sensorial disability; pregnancy and confounding factors for salivary cortisol analysis (the consumption of licorice, tobacco or the practice of physical exercise before the sessions).

### 2.3. Experimental Setup

#### 2.3.1. VR Device

A computer (equipped with a processor Intel Xeon E5-1360 v4 and a graphic card NVIDIA GeForce GTX 1060) was connected to the HTC Vive CV1 system composed of a helmet, two Vive controllers, and two Vive Trackers attached to the feet of the participants whose position and orientation were detected via infrared by two base stations in a space of 4 m × 3 m × 2.5 m (Figure 3). Steam VR and Xsplit software were used to run the environment and to visualize and record the user’s behavior. A second screen was used by the experimenter to follow what was happening in the VR environment from the participant’s point of view.

#### 2.3.2. VR Environment

A multidisciplinary team of physicians and computer scientists co-designed an original exposition environment dedicated to the experimentation using the game engine Unity 3D 5.6. 0f3 (64-bit). The originality of this environment is its fully embodied and dynamic nature (a video of a first-person view is provided in Appendix A). The environment consisted of a 150 m skyscraper in construction with two elevators, with transparent doors leading to the top of the building.

##### Assessment Sessions (V1 and V4)

Firstly, participants were exposed to the ground in the VR without any height situations for 2 min in order for them to get used to the VR equipment and the VR environment, to assess the tolerance and to learn the SUD notation technique by eye tracking. Participants then performed a virtual task for 10 min consisting of riding an open elevator. Starting from the ground, the diagnosis scenario only used one elevator on one side of the building. Participants were invited to get into the elevator, and they were asked to rise as high as possible. On each of the floors (1, 3, 5, 13, 22 and 35), the door opened, and participants had to observe their environment for 30 s before rating the SUD twice with a 30 s interval. SUD assessments were necessary before accessing the next floor by pressing a button provided for this purpose (through gaze-based interaction) (Figure 4). The SUD score was assessed using a visual analog scale ranging from 0 to 10, represented by a gauge on one wall of the elevator (Figure 4 and Appendix A). The participant had to look at the gauge at the desired score from 0 to 10 (0/10 corresponding to no anxiety and 10/10 corresponding to the highest level of anxiety). Once the target was placed on the desired number, the participant had to maintain a fixed gaze for 4 s (eye tracking) in order to record the score.

##### Exposure Sessions (V2 and V3)

During the exposure scenario, we used two elevators, and the virtual task lasted 20 min. Starting from the ground, participants were invited to get into the elevator and were asked to rise as high as possible. At each floor (1, 3, 5, 13, 22, 35, 55, 70 and 99, for a total of 9 levels and a 150 m maximum height), the door opened, and participants had to cross a wooden plank between two platforms in order to access the elevator of the second part of the skyscraper and be able to go up to the next floor (Figure 5). Once the board was crossed, participants had to observe their environment for 30 s before rating the SUD twice with a 30 s interval between both. The first SUD was retained for analysis (corresponding to anxiety increase), while the second SUD score had to display a reduction in anxiety (<4 or divided by two) to ensure the participant’s safe progression. SUD assessments were necessary to access the next floor by pressing a button through eye fixation. Another unity program was developed to calibrate the real plank position, to launch one of the scenarios depending on the session and to record all the events and measurements. To accentuate the realism, participants walked on a real wood plank placed on the floor matching the position of the footbridges in the VR scenario. It was pliant to walk on, and participants’ feet were tracked and displayed in VR to give visuo-haptic feedback and proprioceptive sensation (Figure 3).

As this was a feasibility study, and in order to allow the replication of our work, the details of the virtual reality and electric stimulation procedure are described in the Appendix A.

#### 2.3.3. tDCS Device and Protocol

tDCS was performed using a Starstim^®^ device (Neuroelectrics, Barcelona, Spain). NIC 2.0 software ran on another computer, which enabled us to command the stimulation with sham and double-blind modalities throughout a Bluetooth wireless system. Participants in the active tDCS group received 1 mA anodal stimulation during 20 min, whereas participants from the control group received sham stimulation (30 s of stimulation ramp-on and ramp-up). The stimulation was applied during the 20 min of VR exposure (the computerized activation of NIC 2.0 automatically occurred when the VR software ran the exposure). The anode (Pistim^®^ of π cm^2^) allowed the use of a highly conductive saline gel (Electrode Gel Signagel^®^) and not a wet sponge in order to avoid the humidification of the VR helmet just below, which was placed (and fixed in the head neoprene cap) over FpZ according to the EEG 10–20 system in order to target the ventromedial prefrontal cortex (vmPFC), and the cathode (Sponstim^®^, 25 cm^2^) was placed under the chin according to one previous current direction modelization for this target using an anodal stimulation of 1.5 mA with a 3 × 3 cm^2^ electrode on FpZ and a 5 × 5 cm^2^ return electrode, as published by Junghofer et al. in 2017 [42]; this method was used in a recent tDCS-MEG study demonstrating that vmPFC hypoactivity is associated with fear generalization [47] (Figure 6). The cathode was maintained thanks to a fixation located at the level of the bandoliers of the Starstim cap, which passed under the chin.

### 2.4. Measures

The level of acrophobia, visual height intolerance and anxiety of the participants were evaluated at baseline before the first visit and exposition (V1) and at endpoint after the last exposition during the fourth visit (V4) through several questionnaires and scales: the *Acrophobia Questionnaire* (AQ) (divided in two subscales, anxiety and avoidance) [45], the *Attitude Towards Heights Questionnaire* (ATHQ) [48], the *Height Interpretation Questionnaire* (HIQ) [49], the *Visual Height Intolerance questionnaire* (vHIQ) [12], the *State-Trait Anxiety* questionnaire (STAI-Y-A and STAY-Y-B) [50], the *Clinical Global Impression* (CGI) [51] and the *Subjective Units of Distress* (SUD) [46] were assessed during the exposition at different height degrees. The higher the score on those scales, the higher the disturbance of the subject. The sense of presence experienced in the virtual environment was evaluated with the *Igroup Presence Questionnaire* (IPQ) [52], and the tolerance to VR was evaluated through the *Simulator Sickness Questionnaire* (SSQ) [53] after VRET sessions during the second (V2) and the third (V3) sessions. The tolerance to tDCS was evaluated through a collection of adverse effects perceived by the participants and dropouts rates.

Stress reactivity was also evaluated using cortisol saliva sampling with a Cortisol-Salivette^®^ (Sarstedt, Nümbrecht, Germany) before and after exposition at baseline and at endpoint (V1 and V4). This sampling (2 min in mouth) was realized in the same room, at the same hour for each subject to consider nycthemeral variations in cortisol secretion, and was then collected and sent to the Biochemistry Department of Nantes University Hospital. Salivary cortisol was collected and stored at −80 °C until analysis. Salivary cortisol level was determined using high performance liquid chromatography–tandem mass spectrometry (3200 Qtrap^®^, Sciex, Villebon sur Yvette, France) after extraction from 500 µL of saliva using the liquid–liquid mode with the dichloromethane quantification transition of 363 > 121 and a deuterium internal standard containing cortisol-d4 (Biovalley, Illkirch-Graffenstaden; France). The saliva cortisol rate was measured before and after the sessions (just after and 15 min after).

### 2.5. Ethic

All participants gave written informed consent. All procedures were approved by the Ethics Committee CPP Ile de France 1 (reference: 2017—oct.14679 ND, 06/11/2017).

### 2.6. Statistical Analysis

This is a “proof of concept” pilot study with no a priori knowledge with regard to the hypotheses required for determining the number of subjects needed. Nevertheless, the seminal work of Abend et al. [34] on fear extinction included 30 subjects with 15 subjects for the tDCS condition and 15 for the sham condition. They based their sample on previous research that studied the effects of cannabinoids on mnesic consolidation (28 subjects) [54]. A controlled clinical trial demonstrated the superiority of D-Cycloserine versus placebo when associated with 2 VR sessions for acrophobia with a sample of 27 subjects [20]. Thus, we decided to constitute a convenience sample of 28 subjects.

A descriptive analysis at baseline was conducted where quantitative variables were presented as means and standard deviations (SD), and qualitative variables were presented as numbers and percentages. Non-parametric tests were used to compare both groups: Mann–Whitney’s test for quantitative data and Chi-2 and Fisher’s tests for qualitative data. The mean values, 95% confidence interval and effect size of the difference were provided for each comparison. Missing data are described in terms of frequency for each group, and potential imbalance was evaluated using Fisher’s exact test. For each score, individual changes between V1 and V4 were compared between groups using Mann–Whitney’s test. The mean values, 95% confidence interval and effect size of the difference were provided for each comparison. Analyses were performed with SAS software version 9.4 (SAS Institute, Cary, NC, USA) and R software version 4.1.1 (R Foundation, www.r-project.org).

## 3. Results

Socio-demographic and clinical data at baseline are summarized in Table 1, and the two groups of participants (active/sham) were well balanced.

Our study confirms the technical feasibility of the tDCS + VRET association, and no significant adverse effect was reported.

The mean scores of the sense of the presence in the Igroup Presence Questionnaire (IPQ) at the end of therapeutic exposure sessions were 10.27 ± 6.72 (mean ± standard deviation) in the active group and 6.50 ± 9.42 in the sham group, without between-group significant difference using the Mann–Whitney test (z = −0.960, *p* = 0.36). The tolerance as measured by the SSQ scale at the second (V2) and the third (V3) are shown in Table 2. Of the 28 participants included in the study, 25 completed the study, which resulted in a retention rate of 89.29%. Two subjects could not attend the last VRET session for personal organizational reasons, and one preferred not to complete the protocol due to apprehension of having a headache. Two dropouts occurred in the active group, and there was one dropout in the sham group.

For the whole sample, we observed a significant effect of the height on the SUD scores with a mean increase per floor of 0.31 (95% CI: 0.27, 0.36; *p* < 0.0001). Between-group comparisons did not demonstrate a significant effect of active versus sham stimulations on SUD scores at each floor within VR sessions, including comparison between the first (V1) and the last (V4) session (Figure 7). At V4, there were no significant differences between groups in all variables, as shown in Table 3. However, there was a significant decrease in SUD scores with regard to mean individual changes between V1 and V4 (mean V1 = 8.44 (95% CI: 7.64, 9.24); mean V4 = 6.36 (95% CI: 5.16, 7.56)) in the whole sample (Wilcoxon test, *p* = 0.0002).

There were no significant changes in clinical scale scores between V1 and V4 between the active and sham groups (*p* > 0.05) (Figure 8 and Table 4, including CGI total scores (*p* > 0.1). Concerning the analyses about the maximum floor reached by the participants, we did not find any significant differences between groups at V1 (active: 17.27 ± 7.09; sham: 17.64 ± 6.58; *p* = 0.95) and V4 (active: 28.27 ± 8.14; sham: 25.43 ± 10.00; *p* = 0.53). Data on the time spent on each floor (in seconds) during the different visits did not show any significant difference between the two groups.

## 4. Discussion

To our knowledge, this is the largest proof of concept study regarding anxiety using online, simultaneous, wireless tDCS and a fully active and immersive VR task. Our study confirms the technical feasibility and the acceptability of the association between wireless tDCS and fully embodied, 360° virtual reality exposure with a good tolerance, as reflected by the low dropout rate and low cybersickness scores. We found an effect size of 0.3625 and 0.3582 at V2 and V3, respectively, concerning the difference of tolerance assessed via the SSQ between the active and sham group (in the direction of active stimulation), which is in favor of good tolerance of VR + tDCS in this context.

Future studies may use self-reported questionnaires in addition to the systematic notification of tDCS adverse events by the examiner.

This acceptability is of importance since research dealing with tDCS and VR for anxiety disorders is a promising emerging field. Indeed, VR is a powerful tool to engage the subject in an ecological task, modifying brain activity and thus putative online tDCS effects according to the state-dependence hypothesis, from a driven neurorehabilitation perspective [39,40,41].

The technical feasibility was first demonstrated by the absence of a decrease in the participants’ sense of presence with tDCS (due to its weight or the sensations on the scalp, for instance). Furthermore, no interferences were detected between wireless tDCS and VR using different Bluetooth signals and software. The quality and fluidity of the images in VR and the continuity of tDCS stimulation were not hindered. We also demonstrated that it was feasible to synchronize the two systems with computer programming. Subjects’ movements did not impair the positioning of the different devices and did not alter stimulation impedance, for example. These observations pave the way for further studies exploring potential synergies between two systems in new experimental paradigms.

Our virtual, experimental environment appears validated with an increase in anxiety as a function of the floors climbed by the participants. Moreover, only two VRET sessions clearly improved visual height intolerance compared to the baseline with the use of this specific environment.

First, the involvement of body movements with real proprioceptive sensations (movements of the plank under the feet) and a complete embodiment may be particularly suitable for high visual intolerance management since anxiety and body movement regulation have close reciprocal interactions during exposition to height. Indeed, it has been shown that self-motion has an important role in triggering a fear of heights, since acrophobic participants experience higher levels of anxiety when required to move at a fixed height [55,56]. Future study designs should take into account novel outcomes (e.g., accelerations, pauses, body positions and step length) to assess the effect of tDCS on fear and posture interactions. Nonetheless, one must keep in mind that there are still conflicting results regarding the effect of tDCS on posture [55,56,57,58].

Secondly, a strong sense of presence was reported by the participants. Sense of presence refers to the feeling of being in the virtual world as if you were in the real world and is required to induce emotion and for optimal therapeutic effects in VR [59,60]. Moreover, the activation of the medial prefrontal cortex depends on the sense of presence [61]. Herrmann et al. [30] did not find any effect of rTMS applied to the medial prefrontal cortex on the sensation of presence despite a therapeutic effect on acrophobia ratings. They then hypothesized that the therapeutic effects of rTMS were not mediated by a NIBS-modulated sensation of presence. Our results are in accordance with such a hypothesis, with no intergroup difference in sensation of presence ratings.

Overall, this preliminary study failed to demonstrate an added value of active tDCS. One strength was, therefore, the online stimulation. However, this may be the consequence of several limitations: technical and methodological.

At the methodological level, it is important to consider this work as an initial experiment to understand the effect sizes for this specific design and keep in mind that in general, a between-subject design requires at least between 30 and 40 subjects in tDCS studies [62]. The small size effect (0.1 for the SUD criteria at V4) indicates that the combined VR-tDCS stimulation paradigm must be improved with fundamental studies before the hypothesis of a 0.4–0.5 effect size can be reached. Indeed, the power associated with a 0.1 effect size in this sample would be 5 %, which is low. We calculate that the sample size required to evaluate tDCS on the a priori hypothesis of a medium effect (effect size = 0.5), which seems reasonable for tDCS studies [63], with a power of 80% and a risk of type I error of 5%, would be 63 per arm, 126 in all. The interpretations of *p*-value should therefore be considered very cautiously here. Moreover, the use of a simple randomization in this limited sample constitutes another limitation, as it may lead to suboptimal matching. However, baseline performances seemed to be comparable in both groups.

At the technical level, concerning tDCS parameters, a higher number of sessions could be a way to improve the effects of tDCS. In comparison, on average, the number of tDCS sessions applied on the dorsolateral prefrontal cortex in depression is 15 sessions over a period of 20 to 30 min per session [64,65]. Concerning targeting, the literature was very scarce in 2016 on the optimal tDCS montage for fear inhibition and is still very limited. The orientation of the electric field, which is defined by the polarity and location of the electrodes, influences the effects of tDCS [66]. van’t Wout-Frank and colleagues [36] located the anode regarding AF3, and the cathode was placed facing PO8 according to the international 10–20 EEG system. We chose another kind of stimulation (anode on FpZ and cathode under the chin) that avoid the cathodal modulation of other cortical areas and allowed strong stimulation of the anterior vmPFC in previous modeling. Alternative montages have been proposed to target the vmPFC: anodal AF3 stimulation with cathodal contralateral mastoid to stimulate the left vmPFC [67], concomitant F7 anodal and F8 cathodal stimulation [35] or anodal FpZ and cathodal OZ stimulation [68]. Ultimately, it is not certain that 1 mA stimulation with a smaller anodal electrode (as a precaution in this pilot study, rather than larger electrode with 2 mA) was sufficient to adequately modulate vmPFC activity. An alternate explanation for the absence of superiority with active tDCS may be that a mild additional effect of tDCS may have not been detected since the virtual exposure by itself may have had a strong and fast effect on anxiety. Moreover, our study involved healthy subjects with visual height intolerance and not acrophobia. The neurophysiological rationale, involving hypoactivation of the vmPFC and hyperactivity of the amygdala, was based primarily on the population of patients suffering from major anxiety disorders. We hypothesize that in vIH, these abnormalities may be less pronounced, and it would therefore be more difficult to demonstrate the interest of tDCS in this group of patients in contrast to acrophobia. However, further studies using tDCS are needed, since tDCS showed abilities to modulate cognitive performance in healthy subjects, especially when used *online* rather than *offline* [57,58].

Concerning the objective biological salivary cortisol measurements, our results did not show differential effects in both groups. Most of the studies focused on left dorsolateral prefrontal cortex stimulation and suggested rTMS [69,70] or tDCS [71] could reduce the neuroendocrine response to stress in healthy subjects. Nevertheless, recently, a study evaluating a single session of tDCS on the left dlPFC in healthy subjects showed that tDCS was probably responsible for an attenuation of the autonomic response to stress (heart rate variability) without a significant reduction in cortisol concentration [72]. To our knowledge, no study investigated the effect of tDCS targeting the vmPFC on autonomic and neuroendocrine responses to stress, whereas it is involved in stress reactivity [73]. It is not known to what extent tDCS could enhance or counterbalance cortisol-lowering associated or not with VR exposure. Heart rate variability and galvanic skin response may also be considered as an outcome in further studies. Ultimately, it is not well known as to when cortisol should be ideally measured in this kind of paradigm. Perhaps it would be better to add several measurement points between 20 and 40 min after the session [74].

## 5. Conclusions

Our study constituted a first step to evaluate the feasibility and tolerance of the concomitant use of online tDCS stimulation and wireless, active and fully embodied VR exposure. Systematic fundamental studies are required to provide information about the optimal area (including laterality), duration, intensity, polarity/frequency and timing of stimulation. More fundamental studies using current diffusion modeling or fMRI-compatible tDCS devices will be useful to determine the best tDCS montage before conducting larger studies. This study paves the way for further research for fear extinction in phobias. It also shows the feasibility and acceptability of this combination, creating a new therapeutic perspective since the association of tDCS and VR could be useful for many other psychiatric conditions (e.g., in depression with vmPFC or dlPFC neuromodulation to decrease negative bias toward negative stimuli or ruminations).

## Figures and Tables

**Figure 1 jcm-11-00345-f001:**
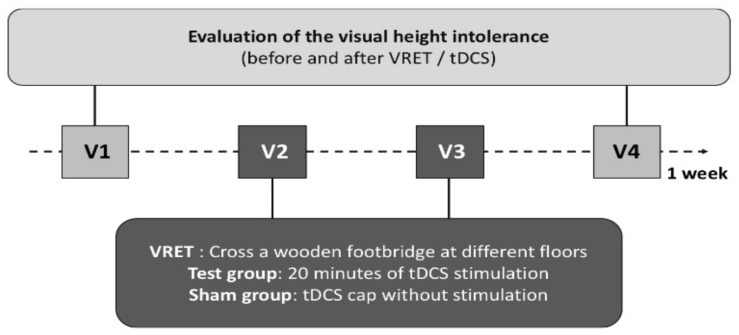
The study including two diagnostic sessions (V1 and V4) and two exposure sessions with tDCS or sham stimulation (V2 and V3).

**Figure 2 jcm-11-00345-f002:**
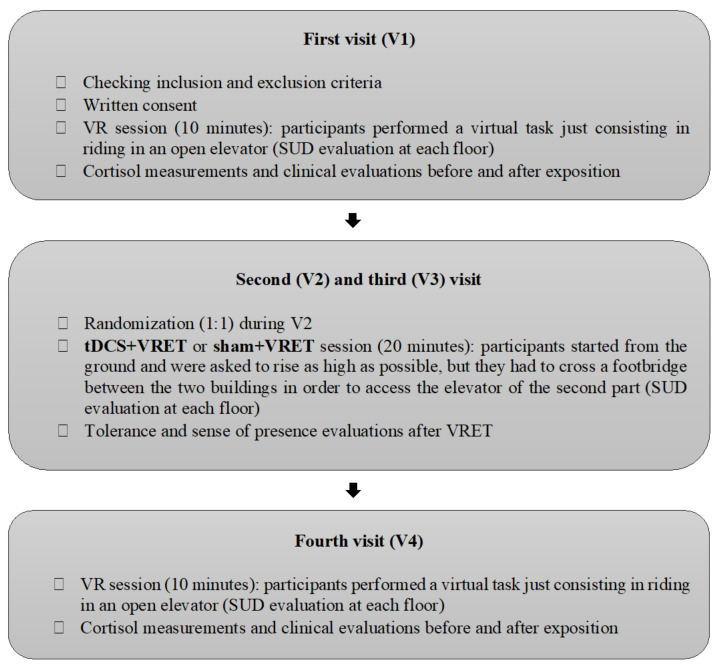
Flow-Chart. Abbreviations: SUD: Subjective Unit of Discomfort; tDCS: transcranial Direct Current Stimulation; VRET: Virtual Reality Exposure Therapy; VR: Virtual Reality.

**Figure 3 jcm-11-00345-f003:**
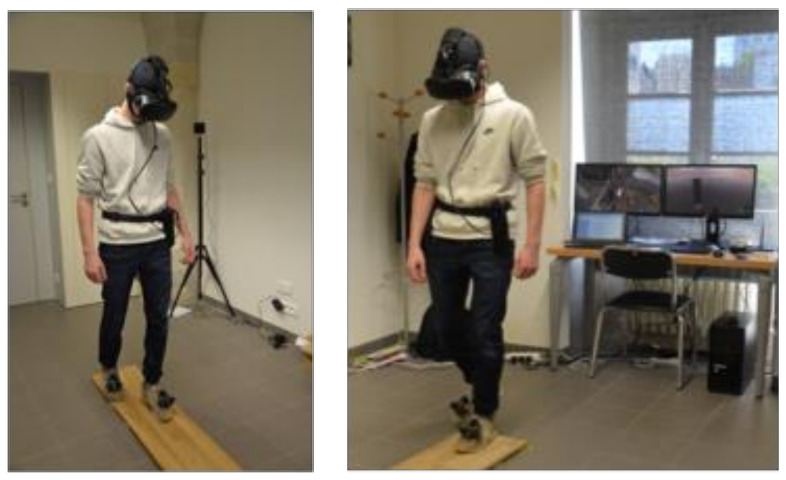
The participant on the plank is equipped with a tDCS cap, a VR helmet and two trackers attached to his feet. The computers in the room provided a 1st and 3rd person view of the participant’s virtual environment in real time.

**Figure 4 jcm-11-00345-f004:**
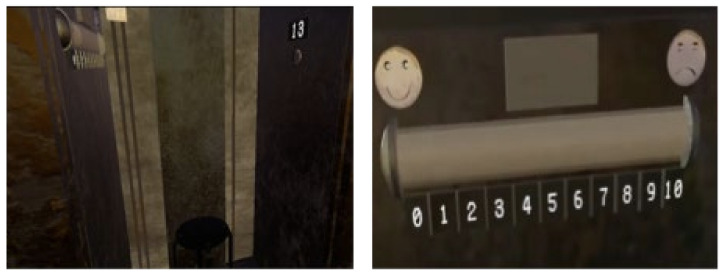
Screenshot of the elevator in the virtual environment (image on the left), including a stool at the bottom of the elevator, the SUD scale on the wall on the left of the image and a button to go to the upper floors on the wall on the right of the image. The detail of the gauge, representing the SUD score from 0 to 10, is on the right of the image.

**Figure 5 jcm-11-00345-f005:**
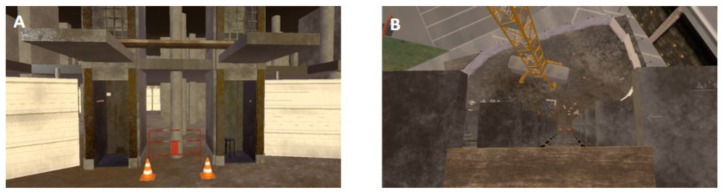
Screenshot of two elevators allow to climb the building (**A**). Cross between two platforms on a plank allows to reach the next elevator in a building with 99 floors (**B**).

**Figure 6 jcm-11-00345-f006:**
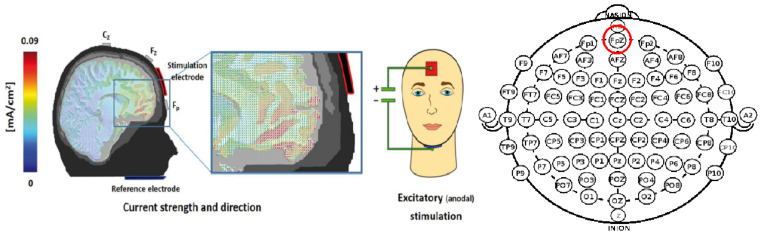
Electrode placement and current modelization. Adapted from the figure published in Junghöfer et al., 2017 [42] with permission of the author.

**Figure 7 jcm-11-00345-f007:**
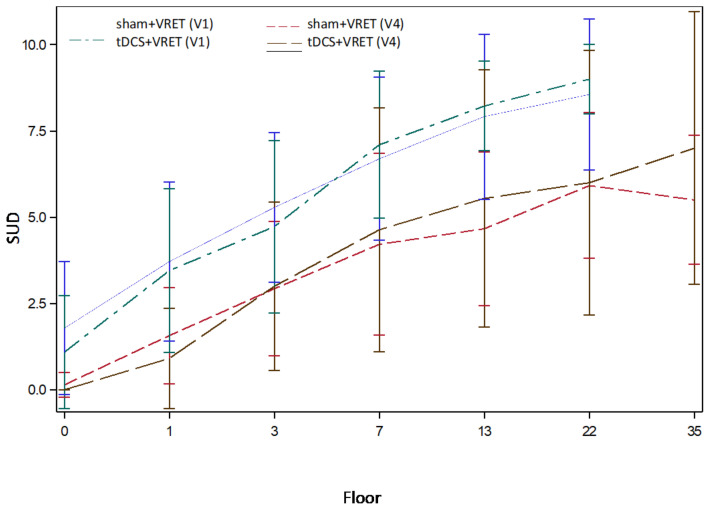
SUD score evolutions according to floor elevations during VRET. Abbreviations: SUD: Subjective Unit of Discomfort; VRET: Virtual Reality Exposure Therapy.

**Figure 8 jcm-11-00345-f008:**
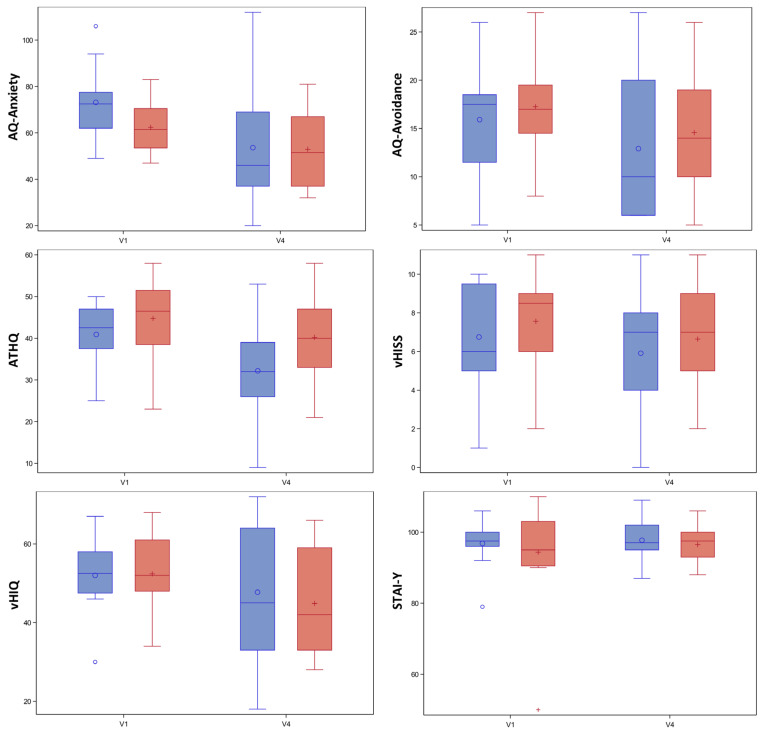
Boxplots of clinical scale scores between V1 and V4. In blue, the tDCS + VRET group (*n* = 11) and in red, the sham + VRET group (*n* = 14). Abbreviations: AQ: Acrophobia Questionnaire; ATHQ: Attitude Towards Heights Questionnaire; STAI: State-Trait Anxiety; vHISS: visual Height Intolerance Severity Scale; vHIQ: visual Height Intolerance Questionnaire; VRET: Virtual Reality Exposure Therapy.

**Table 1 jcm-11-00345-t001:** Socio-demographic and clinical characteristics of study sample at the baseline (V1).

	tDCS + VERT (*n* = 11)	Sham + VERT (*n* = 14)	Stat	*p*-Values
Sex (male)	2 (18.2%)	5 (35.7%)	χ^2^ = 0.939	0.332
Age	37.27 ± 15.88	37.79 ± 17.68	z = −0.164	0.869
AQ anxiety	74.00 ± 15.80	64.14 ± 10.41	z = −1.591	0.112
AQ avoidance	15.73 ± 6.23	17.21 ± 2.94	z = −0.852	0.394
ATHQ	41.36 ± 8.15	45.57 ± 9.80	z = −1.372	0.170
HIQ	52.27 ± 9.80	52.79 ± 9.96	z = −0.302	0.763
STAI-Y-A	48.64 ± 2.01	48.71 ± 4.68	z = −0.277	0.782
STAI-Y-B	49.82 ± 3.09	49.07 ± 4.60	z = −0.826	0.409
STAI-Y total	98.45 ± 3.64	97.79 ± 7.04	z = −0.823	0.410
vHISS	6.82 ± 2.86	7.57 ± 2.85	z = −0.635	0.526
CGI Severity	4.91 ± 0.70	4.79 ± 0.98	z = −0.119	0.905
SUD max	8.55 ± 2.34	8.36 ± 2.27	z = −0.173	0.863

Data are *n* (%) or mean (SD); z = z value for Mann–Whitney U test; χ^2^ = Chi-square test. Abbreviations: AQ: Acrophobia Questionnaire; ATHQ: Attitude Towards Heights Questionnaire; CGI: Clinical Global Impression; HIQ: Height Interpretation Questionnaire; STAI: State-Trait Anxiety; SUD: Subjective Unit of Discomfort; VRET: Virtual Reality Exposure Therapy; vHISS: visual Height Intolerance Severity Scale.

**Table 2 jcm-11-00345-t002:** Simulator Sickness Questionnaire (SSQ) scores at V2 and V3.

	Mean (Sham Group)	95% CI Lower	95% CI Upper	Mean (Active Group)	95% CI Lower	95% CI Upper	Effect Size
SSQ Scale at V2 SSQ scale at V3	8.642 5.000	6.658 3.323	10.628 6.677	11.364 7.909	5.368 1.246	17.359 14.572	0.363 0.358

**Table 3 jcm-11-00345-t003:** Mean difference between groups for each score at V4 (with 95% CI and effect size). Abbreviations: AQ: Acrophobia Questionnaire; ATHQ: Attitude Towards Heights Questionnaire; CGI: Clinical Global Impression; HIQ: Height Interpretation Questionnaire; STAI: State-Trait Anxiety; SUD: Subjective Unit of Discomfort; VRET: Virtual Reality Exposure Therapy; vHISS: visual Height Intolerance Severity Scale.

Scores	Mean (Sham Group)	95% CI Lower	95% CI Upper	Mean (Active Group)	95% CI Lower	95% CI Upper	Effect Size
SUD max	6.2100	4.9800	7.4400	6.5500	4.1500	8.9500	0.1046
AQ anxiety	52.9300	44.4900	61.3700	53.6400	37.6800	69.6000	0.0326
AQ avoidance	14.5700	11.1700	17.9700	12.9100	8.1200	17.7000	−0.2306
ATHQ	40.2100	34.4300	45.9900	32.1800	25.2500	39.1100	−0.7193
vHISS	6.6400	5.2400	8.0400	5.9100	3.8700	7.9500	−0.2410
HIQ	44.8600	37.3200	52.4000	47.7300	37.3200	58.1400	0.1820
STAI	96.5000	93.9776	99.0224	97.7273	94.2837	101.1713	0.2343
CGI total	15.3571	12.9703	17.7440	16.0909	11.6314	20.5505	0.1201
IPQ	6.5000	1.4649	11.535	10.2727	6.2177	14.3273	0.4610
SSQ at V2	8.6429	6.6582	10.627	11.3636	5.3680	17.3592	0.3625
SSQ at V3	5.0000	3.3227	6.6772	7.9090	1.2460	14.5721	0.3582
Cortisol level after exposure	0.9214	0.5945	1.2484	0.8636	0.5442	1.1830	−0.1010

**Table 4 jcm-11-00345-t004:** Mean individual changes between V1 and V4 for each score in both group with 95% CI and effect size (a positive effect in the sense of active tDCS and a negative one in the sense of sham tDCS).

Scores	Mean (Sham Group)	95% CI Lower	95% CI Upper	Mean (Active Group)	95% CI Lower	95% CI Upper	Effect Size	*p*-Value *
SUD max	−2.14	−3.21	−1.07	−2.00	−3.81	−0.19	0.0549	0.484
AQ anxiety	−11.21	−18.16	−4.26	−20.36	−34.83	−5.89	−0.4741	0.228
AQ avoidance	−2.64	−5.31	0.03	−2.82	−7.04	1.40	−0.0296	0.913
ATHQ	−5.36	−9.10	−1.62	−9.18	−17.62	−0.74	−0.3451	0.583
vHISS	−0.93	−1.46	−0.40	−0.91	−2.12	0.30	0.0126	0.955
HIQ total	−7.93	−13.81	−2.05	−4.55	−12.99	3.89	0.2685	0.459
STAI total	−1.29	−4.50	1.92	−0.73	−3.75	2.29	0.1014	0.978
Cortisol level after exposure	−0.29	−0.82	0.24	0.11	0.11	0.11	0.5657	0.131

* non-parametric Kruskal–Wallis rank sum test. Abbreviations: AQ: Acrophobia Questionnaire; ATHQ: Attitude Towards Heights Questionnaire; HIQ: Height Interpretation Questionnaire; STAI: State-Trait Anxiety; SUD: Subjective Unit of Discomfort; vHISS: visual Height Intolerance Severity Scale.

## Data Availability

Data available on reasonable request.

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
