# Peer review of "Feasibility of Combining Transcranial Direct Current Stimulation and Active Fully Embodied Virtual Reality for Visual Height Intolerance: A Double-Blind Randomized Controlled Study"

_jcm, 2022, doi:10.3390/jcm11020345_

Round 1

Reviewer 1 Report

The article by Bulteau and coll. covers a very interesting and poorly explored topic, i.e. the interaction between transcranial stimulation and virtual reality. In their experiment, the authors compared two groups of 14 healthy subjects with significant fear of heights with respect to their response to a combined treatment of virtual reality exposure therapy and active or sham tDCS. The primary outcome was feasibility, which came out to be good. The secondary outcomes were the effects of the protocol on the salivary cortisol and anxiety of participants, which did not yield conclusive information.

I believe that this paper has several strengths and some minor flaws that can be improved with some changes before publication. I hope that the following suggestions might help do it.

Abstract

  • When outlining the methods, the authors should say something about the VRET protocol, otherwise the sentence “a significant increase in anxiety at each level of difficulty” comes unexpected and is not clear which difficulty they are referring to
  • “We observed in two groups”: maybe “both groups” sounds better

Introduction

  • After the first sentence there are two periods back-to-back
  • “This portative easy-to-use device “: maybe “portable” instead of portative sounds better
  • The aims stated at the end of this section are not consistent with those in the abstract. In particular, while in the abstract tolerance was a primary aim (with feasibility), here is secondary.
  • Always in the aims, when the authors say “the efficacy of virtual exposure” should specify efficacy in terms of what (I presume in reducing the height-induced anxiety)

Methods

Design

  • “Two assessment sessions (V1 and V4) before and after the VRET sessions to measure participants’ fear of height thanks to a 10 minutes virtual exposure between baseline and after treatment”. Not very clear…maybe should be rephrased (“following 10 minutes virtual exposure” instead of “thanks to a”?....eliminating “between baseline and after treatment”?)
  • Figure 2:

first box: “without bridges to cross” comes unexpected, can be eliminated

second box: should be VRET instead of VERT (in the figure’s caption as well)

third box: I would put “without bridges to cross” in brackets

Participants

  • “They were considered healthy”: I recommend specifying a little further what you mean by “healthy” … not having a psychiatric disorder?
  • “and a Subjective Unit of Distress (SUD) rating ≥ 5/10 [48]”: 1) the reference doesn’t seem to be related to the concept of SUD… maybe the suitable is the 53 here; 2) it is not clear to me why the subject should have a SUD ≥ 5/10 at baseline to enter the study, this point should be clarified

VR environment

  • “(Video 1 first-person view, and Video 2 third-person for third person view in supplementary materials)”: not available
  • “to learn the SUD notation technique by eye tracking”: the authors could explain in a little more detail what a SUD is and how the participants rated it
  • “Once the board was crossed, participants had to observe their environment for 30 seconds before rating the SUD twice with a 30-second interval”: the authors could explain why they chose to make the participants rate the SUD after crossing the board and not before (or both before and after)

tDCS device and protocol

  • It would be nice to know more about the practical aspects of stimulation, so that other researhers could replicate the experiment. Electrodes dimensions? How to fix the reference electrode under the chin? (which is not reported in the reference cited (43)

Measures

  • My concern is that not all the rating scales can be used in healthy subjects…for example as far as I know the CGI is only for a clinical population
  • As concerns feasibility, reporting a retention rate measure (percentage of subjects that terminated the protocol among those who started) could be useful

Statisitcal analysis

  • “Alpha risk for significance was set at 5%”: Being a pilot study without a proper sample size calculation (and therefore probably underpowered), I do not recommend to establish a threshold for statistical significance, due to the high risk of type I error. Moreover, when reporting p-values, a disclaimer should be always added that the study is not adequately powered and p-values should therefore be considered very cautiously.

Results

  • “we did not find any significant differences between groups at V1 (active: 17.27 ± 7.09; sham: 17.64 ± 6.58; p = 0.95) and V4 (active: 27 ± 8.14; sham: 25.43 ± 10.00; p = 0.53)”. Consistently with my previous comment on statistics, this pessimistic interpretation of the result regarding active tDCS could be revised. In fact, in a pilot study like this, p-values are less important than estimations and confidence intervals to indicate the size and direction of the intervention’s effect. What is surely important is to establish a minimum clinically important difference in the outcome measure. (see Lee et al. BMC Medical Research Methodology 2014, 14:41). After all, if I understand the results, the subjects receiving active tDCS managed to go 3 floors (on average) higher than the subject of the sham group. If the authors believe that this difference of three floors might have some clinical importance, they could (cautiously) underline it.
  • Figure 8 caption: VRET and not TERV

Discussion

  • I would remove the bullets, and the sentence: “This may be due to several characteristics”:
  • P08 should be PO8
  • “Concerning the objective biological salivary cortisol measurements,…..” . Given the acknowledged uncertainty about the neurophysiological effect of the specific tDCS montage/protocol used, the authors should also hypothesize that active tDCS reversed/counterbalanced the cortisol-lowering effect of VRET seen in the sham group.

Reviewer 2 Report

This is overall a nice study showing the feasibility of the combined tDCS-VRET intervention for anxiety.  Though I am fully supportive of publishing both negative and positive results - especially in the tDCS research where the sample sizes are small and most of the studies are strictly speaking underpowered, here I believe it is important to highlight that the study did not find significant tDCS effect.  My worry is that the authors framed the papers as a feasibility study only after discovering it had null effects. Therefore I would recommend a much more balanced representation of the study aims and objectives. Also, the sentence "significant reduction of anxiety after the stimulation" is misleading as it suggests that there was a significant effect of stimulation, which is not the case. 

The introduction is nicely written and covers all points relevant to the presented work. 

Most of my comments will focus on Methods and Results since that is my primary area of expertise. 

In the Design, it should be explicitly stated that this was a between-subjects design and the randomization procedure needs to be presented in more detail.  As simple randomization can be inappropriate for small samples, I strongly encourage authors to discuss the limitations of the study that stem from this aspect of the method.

How was the sample size determined? The statistical analysis list 2 previous studies with 30 and "only 27" participants, it remains unclear to me why did the authors opt for 28? Why is the number of participants unequal across the groups? Were any subjects excluded? The good practice is to do the prior power analysis to estimate the number of participants needed to detect a certain effect size. If unsure about the exact effect size a more conservative estimation should be adopted. I suggest that authors either report on the power analysis or perform post-hoc sensitivity analysis. This study is clearly underpowered but most of the tDCS studies are, so I believe authors need to address this issue in the methods and discussion section. 

I suggest the following papers as relevant references addressing the design and the sample issues:

  • https://www.ncbi.nlm.nih.gov/pmc/articles/PMC5020062/
  • https://pubmed.ncbi.nlm.nih.gov/34605816/
  • https://www.ncbi.nlm.nih.gov/pmc/articles/PMC5997279

In the description of the tDCS procedure, some important information is missing: What was the size of the electrodes? Current density? Was sham double or single rump-up/down? What is "Nikon" the after NIC 2.0? I suggest authors provide the Figure of electric field molding instead of Figure 6 (it can be generated from NIC or any other free software for field molding such as STIMNIBS, ROAST, COMETS), as this is a slightly unusual electrode montage. Also, It is unclear to me when the stimulation was applied with respect to the task? Why are there 2 sessions and not one or 3? I am missing the rationale behind this decision. 

In statistical analysis it seems that authors are switching between parametric and non-parmatric tests -  why is that the case?

Since the paper claims its primary purpose is to evaluate feasibility - that is the lack of adverse effects, it remains unclear to me how this primary outcome was measured? was there a questionnaire for self-report of adverse effects? The results just state "no adverse effects were reported" but no data was provided to support this claim. 

I would suggest adding the column with statistics and exact significance (confidence intervals are welcome as well) in Table 1 so all differences are reported. The text does not need to repeat the content of the table. 

Minor suggestions:

The first sentence under 2.3.2. has oddly capitalized word "Doctor". There is no need to capitalize it, the same as the Computer Scientist is not capitalized. regardless of the capitalization, I believe that this part of the sentence should be omitted as it is not relevant who designed the environment, but how it was designed (the fact that someone is MD or CS is in no way guarantee of the adequate design)

- "rump on" should be changed to "rump up"

Author Response

Reviewer #2 :

This is overall a nice study showing the feasibility of the combined tDCS-VRET intervention for anxiety.  Though I am fully supportive of publishing both negative and positive results - especially in the tDCS research where the sample sizes are small and most of the studies are strictly speaking underpowered, here I believe it is important to highlight that the study did not find significant tDCS effect.  My worry is that the authors framed the papers as a feasibility study only after discovering it had null effects. Therefore I would recommend a much more balanced representation of the study aims and objectives. Also, the sentence "significant reduction of anxiety after the stimulation" is misleading as it suggests that there was a significant effect of stimulation, which is not the case. The introduction is nicely written and covers all points relevant to the presented work. Most of my comments will focus on Methods and Results since that is my primary area of expertise. In the Design, it should be explicitly stated that this was a between-subjects design and the randomization procedure needs to be presented in more detail.  As simple randomization can be inappropriate for small samples, I strongly encourage authors to discuss the limitations of the study that stem from this aspect of the method.

We thank the reviewer for his positive comments and constructive feedback. This was indeed a feasibility study, but we have to admit that following your remarks, we put the feasibility aspects, especially the technical elements, in the background during the drafting process to the detriment of the results of the clinical and biological measurements. In order to put the emphasis back on the feasibility elements resulting from several years of work, we have made some modifications to the manuscript.

  • First, the abstract is more enriched on the feasibility elements.
  • In the last paragraph of the introduction : “We hypothesized that it may be technically feasible to associate wireless online tDCS with VRET for vHI using an active fully embodied and immersive  virtual task. Our primary outcomes for feasibility were the sense of presence, cybersickness and dropout rates. The second aim of this initial experiment was to describe the efficacy (and understand the effect size with this particular experimental setting) of the virtual exposure with and without add-on tDCS in reducing height-induced anxiety.”

  • Figures 3 and 4 have been enriched with photos to better understand the modality of our study
  • We have added in the supplementary material a video of the VRET (Supp Material 1) and a step by step description of the protocol and the launching of the VRET/tDCS in order to provide the necessary elements to replicate some aspects of our work (Supp Material 2).
  • In the last paragraph of the discussion we have added the following sentence: The technical feasibility was first demonstrated by the absence of decrease in the participant's sense of presence with tDCS (due to its weight or the sensations on the scalp for instance). Furthermore,  no interferences were detected between wireless tDCS and VR using different bluetooth signal and softwares. Quality and fluidity  of images in VR, continuity of tDCS stimulation were not hindered. We also demonstrated that it was feasible to synchronize the two systems with computer programming. Subject’s movements didn’t impaired positioning of the different devices and didn’t alter stimulation impedance for example. Those observations pave the way for further studies exploring potential synergies between two systems in new experimental paradigms. “      
  • We retrieved the sentence "significant reduction of anxiety after the stimulation" as
  • We precise paragraph 2.1 the between-subject comparison as follow : “ This study was conducted using a randomized, double-blind, and controlled between-subjects design”
  • We added in the discussion the limitation concerning the use of a simple randomization in this small sample : “At the methodological level, it is important to consider this work as an initial experiment to understand the effect sizes for this specific design and keep in mind that a between-subject design requires in general at least between 30 and 40 subjects in tDCS studies [62]. The small size effect (0.1 for the SUD criteria at V4) indicates that combined VR-tDCS stimulation paradigm must be improved with fundamental studies before being able to reach the hypothesis of a 0.4-0.5 effect size. Indeed, the power associated with a 0.1 effect size in this sample  would be  5 % which is low. We calculate that the sample size required to evaluate tDCS on the a priori hypothesis of a medium effect (effect size= 0.5), which seems reasonable for tDCS studies [63],  with a power of 80% and a risk of type I error of 5%, would be 63 par arm, 126 in all.  The interpretations of p-value should therefore be considered very cautiously here. “

How was the sample size determined? The statistical analysis lists 2 previous studies with 30 and "only 27" participants, it remains unclear to me why did the authors opt for 28? Why is the number of participants unequal across the groups? Were any subjects excluded? The good practice is to do the prior power analysis to estimate the number of participants needed to detect a certain effect size. If unsure about the exact effect size a more conservative estimation should be adopted. I suggest that authors either report on the power analysis or perform post-hoc sensitivity analysis. This study is clearly underpowered but most of the tDCS studies are, so I believe authors need to address this issue in the methods and discussion section.  I suggest the following papers as relevant references addressing the design and the sample issues:

  • https://www.ncbi.nlm.nih.gov/pmc/articles/PMC5020062/
  • https://pubmed.ncbi.nlm.nih.gov/34605816/
  • https://www.ncbi.nlm.nih.gov/pmc/articles/PMC5997279

The referee is right. We add in the discussion  that we should consider this experiment as an  initial experiment to understand the effect sizes for this specific design and keep in mind that a between subject design requires in general between 30 and 40 subjects in tDCS studies ((Bjekić, J., Živanović, M., Filipović, S. R. Transcranial Direct Current Stimulation (tDCS) for Memory Enhancement. J. Vis. Exp (175), e62681, doi:10.3791/62681 (2021). Moreover, we calculate that The sample size required to evaluate a medium effect size, with a power of  and a risk of type I error of , is  par arm,  in all. We presented effect size in Table 3 and 4 (please see above) and the small size effect (0.1 for the SUD criteria at V4) indicates that combined VR-tDCS stimulation paradigm must be improved with fundamental studies before being able to reach the hypothesis of a 0.4-0.5 effect size. Indeed, the power associated with a 0.1 effect size in this sample  would be  5 % which is low. However, we found an effect size of 0.3625 and 0.3582 at V2 and V3  respectively for the effect size of the difference of tolerance assessed by SSQ between active and sham group (in the direction of active stimulation) which is in favor of a good tolerance of VR in this context.

In the description of the tDCS procedure, some important information is missing: What was the size of the electrodes? Current density? Was sham double or single rump-up/down? What is "Nikon" the after NIC 2.0? I suggest authors provide the Figure of electric field molding instead of Figure 6 (it can be generated from NIC or any other free software for field molding such as STIMNIBS, ROAST, COMETS), as this is a slightly unusual electrode montage. Also, It is unclear to me when the stimulation was applied with respect to the task?

The reviewer draws important remarks. We addressed all the comment in this paragraph in method section : “ The tDCS was performed using a Starstim® device (Neuroelectrics, Barcelona, Spain). The NIC 2.0 software was  running on another computer enabled to command the stimulation with sham and double-blind modalities throughout a Bluetooth wireless system. Participants in the active tDCS group received a 1mA anodal stimulation during 20 minutes, whereas participants from the control group received sham stimulation (30 seconds stimulation ramp-on and ramp-up). The stimulation was applied during the 20 min of VR exposure (A computerized activation of NIC 2.0 automatically occured when the VR software ran the exposure). The anode  (Pistim® of π cm²) allowing the use of a  highly conductive saline gel (Electrode Gel Signagel®)  and not a wet sponge, in order to avoid a humidification of VR helmet placed just below was placed (and fixed in the head neoprene cap) over FpZ according to EEG 10-20 system in order to target the ventromedial prefrontal cortex (vmPFC), and the cathode (Sponstim®, 25 cm2) was placed under the chin according to one previous current direction modelization for this target using an anodal stimulation of 1.5 mA with a 3x3 cm2 electrode on FpZ and a 5x5 cm2 return electrode as published by Junghofer et al. in 2017 [42] and used in a recent tDCS-MEG study demonstrating that vmPFC hypoactivity is associated with  fear generalization  [47] (Figure 6). The cathode was maintained thanks to a fixation located at the level of the bandoliers of the Starstim cap which passed under the chin.”

We explain more in the discussion the pro and cons of such a montage and describe alternative montage even if literature is still very scarce about tDCS over vmPFC for anxiety (5 references only in Pubmed) : “At the technical level, concerning tDCS parameters, a higher number of sessions could be a way to improve tDCS effect. In comparison, the number of tDCS sessions applied on the dorsolateral prefrontal cortex in depression is on average between 5 to 15 sessions over a period of 20 to 30 minutes per session [64][65]. Concerning targeting literature was very scarce in 2016  on optimal tDCS montage for fear inhibition, and is still very limited. The orientation of the electric field, which is defined by the polarity and location of the electrodes, influences the effects of tDCS [66]. van’t Wout-Frank’s and colleagues[67] located the anode regarding AF3 and the cathode was placed facing PO8 according to the international 10-20 EEG system. We chose another kind of stimulation (anode on FpZ and cathode under the chin) that can also be considered “quasi” reference free (avoiding cathodal modulation of other cortical areas) and that has revealed strong stimulation of the anterior vmPFC in previous modeling. Alternative montage have been proposed to target vmPFC : anodal AF3 stimulation with cathodal contralateral mastoid to stimulate left vmPFC [68], concomitant F7 anodal and F8 cathodal stimulation [69] or anodal FpZ and cathodal OZ stimulation [70]. Eventually, it is not certain that 1 mA stimulation with a smaller anodal electrode (as a precaution, rather than larger electrode with 2mA) had been sufficient to adequately modulate vmPFC activity. An alternate explanation for the absence of superiority with active tDCS may be that a mild additional effect of tDCS may have not been detected since the virtual exposure by itself may have had a strong and fast effect on anxiety.”

Why are there 2 sessions and not one or 3? I am missing the rationale behind this decision. 

We have chosen 2 sessions since  exposure therapy in acrophobia requires at least 2 sessions in the literature and as 2 sessions of augmented VR by D–Cycloserine seemed to efficient to reduced fear of height (ref n°20). The therapeutic aim with add-on tDCS and active fully embodied VR is to reach a greater effect with fewer sessions, that’s why we focused on a shorter exposure paradigm which could be  the goal in the  . We justified this point as follow in the text (2.1 Design) : “We chose that number since the aim of research about potentiating the VRET effect with other interventions is to have a stronger and quicker effect [20]”

In statistical analysis it seems that authors are switching between parametric and non-parmatric tests -  why is that the case?

The referee is right. We only kept non-parametric analysis (Mann-Whitney, Wilcoxon, Kruskall-Wallis when appropriate) and didn’t keep the parametric method (linear regression with random effect). We replaced this analysis by the comparison of the differences in mean individual changes between V1 and V4 on the variables of interest as shown in Table 3 and 4. Of note these methodological changes modified the interpretation and conclusion about cortisol levels that were no longer different between groups with this more adequate analysis. We thus retrieve the last Table and reduce the text in the results section.

Since the paper claims its primary purpose is to evaluate feasibility - that is the lack of adverse effects, it remains unclear to me how this primary outcome was measured? was there a questionnaire for self-report of adverse effects? The results just state "no adverse effects were reported" but no data was provided to support this claim. 

The SSQ questionnaire was used to assess VR tolerance. tDCS tolerance was assessed according to the systematic notification of declarative adverse effect by the subject to the examiner. One subject (the one who dropped out, presented with a mild headache  and was anxious about a potential increase in this side effect and preferred to stop the protocol). No self-report for tDCS tolerance was used therefore. We integrated this point as a limitation in discussion.

I would suggest adding the column with statistics and exact significance (confidence intervals are welcome as well) in Table 1 so all differences are reported. The text does not need to repeat the content of the table. 

We added the statistics and exact significance of the baseline characteristics

Minor suggestions:

The first sentence under 2.3.2. has oddly capitalized word "Doctor". There is no need to capitalize it, the same as the Computer Scientist is not capitalized. regardless of the capitalization, I believe that this part of the sentence should be omitted as it is not relevant who designed the environment, but how it was designed (the fact that someone is MD or CS is in no way guarantee of the adequate design)

- "rump on" should be changed to "rump up"

As requested by the reviewer, we have corrected the sentence to: "A multidisciplinary team of physicians and computer scientists" as well as "rump up".

modif

à préciser dans le manuscrit

Round 2

Reviewer 2 Report

I believe authors really put a lot of efforts in adjusting the manuscript, and I highly apprishiate detailed response that mead it easier for me to see the changes that were made. Overall I find the study important  and now as it is written it really shows the effort behind designing an implementing such a complex and technically-challanging designs. Also I am really pleased with how the authors discuss important issues like electrode positioning and sample size -  as this is one of the first papers in this filed I believe it will due to the in depth discussion of the realistic issues have a grate impact on works to come. I fully support the publishing of this paper in its current form.